# Identification and Mitigation of Subsidence and Collapse Hazards in Karstic Areas: A Case Study in Alcalá de Ebro (Spain)

Alberto Gracia [1], Francisco Javier Torrijo [2], Julio Garzón-Roca [3,*], Miguel Pérez-Picallo [1] and Olegario Alonso-Pandavenes [4]

[1]  Associated Technical Consultants, C.T.A., S.A.P., 50006 Zaragoza, Spain; agracia@cta-consultores.com (A.G.); mperez@cta-consultores.com (M.P.-P.)

[2]  Department of Geotechnical Engineering, Research Centre for Architecture, Heritage and Management for Sustainable Development (PEGASO), Universitat Politècnica de València, Camino de Vera s/n, 46022 Valencia, Spain; fratorec@trr.upv.es

[3]  Department of Geodynamics, Stratigraphy and Paleontology, Faculty of Geology, Complutense University of Madrid, 28040 Madrid, Spain

[4]  Geology and Mining Engineering Faculty-FIGEMPA, Central University of Ecuador, Quito 170521, Ecuador; omalonso@uce.edu.ec

*   Correspondence: julgarzo@ucm.es; Tel.: +34-913944864

**Abstract:** Sinkholes are a severe problem in urban areas located in karstic regions, especially where evaporitic rocks such as gypsum exist. Identification and proposal of mitigation measures are needed to reduce this geo-hazard effect on buildings and social urban living. This paper presents a case study of the town of Alcalá de Ebro (Spain), which is located in the highest sinkhole risk region of Europe. The identification and mitigation of a series of sinkholes that appeared are analyzed. The former involves a geological investigation, including boreholes, field tests and geophysics. The latter is addressed by the use of geogrids, mortar injections and polyurethane injections. A complementary finite element analysis is carried out to set the ground behavior associated with the sinking process and assess its future evolution. The Ebro River appears to be the main cause of sinkholes, and results show that ground treatments applied were successful in their purpose, as there are no apparent deformations indicating that the subsidence or sinking process is still active in the area. The use of different techniques depending on the size of the sinkhole, the objectives pursued and the element affected is discussed.

**Keywords:** collapse hazard; sinkhole mitigation; gypsum karst; Alcalá de Ebro; injections; geogrid; evaporitic rocks; finite element analysis





## 1. Introduction

A sinkhole is a type of geo-hazard that essentially consists of a depression or hollow in the ground. This phenomenon is becoming a particularly severe problem in urban areas located in karstic regions [1,2]. Those regions are characterized by the presence of soluble rocks, either carbonate (e.g., limestone) or evaporitic (e.g., gypsum and halite), and cover around 20% of the Earth's ice-free continental surface [3,4]. The dissolution of these soluble strata (rocky substratum) generates a process of upward subsidence towards the surface that leads, in some cases, to episodes of collapse. The sinkhole development process is faster in evaporitic rocks (gypsum is around 100 times more soluble than carbonate rocks, and this ratio is even higher for halite). Thus, the existence of such rocks is a source of problems in built environments [5]. To counteract this hazard, its identification is needed, and mitigation techniques must be applied to reduce its development.

Sinkholes appear randomly due to the gravitational movement of the overlying material found over the soluble rock stratum [1,6,7]. Such overlying material is commonly made

of residual soils, i.e., the in situ disturbed substrate. The gradual dissolution of soluble rocks located at depth, as a consequence of infiltration and passage of water, leads to the formation of cavities or domes near the contact between residual soils and soluble rock [8]. As a consequence, inverse strength profiles are typical of karstic areas, i.e., a reduction in shear strength is observed with an increase in depth (usually, residual soils increase their shear strength with depth).

Hydrological processes, groundwater recharge and rainfall have a critical influence on sinkholes development rate [1,9]. Although karstic materials can remain stable for a long time (and can withstand construction loads with safety), seasonal and forced variations in the water level easily lead to the softening and erosion of the overlying materials. That results in a gradual and progressive formation of domes of disturbed soils. Particularly, water table decline is the human-induced effect that mostly affects sinkhole development [2,10]. These aspects may be aggravated by the current climate change. Construction activities may also change the surface hydrology or reduce the thickness of soils above potential cavities, thus having a great influence on sinkhole development.

The study of the formation processes of sinkholes and their evolution has been carried out from different approaches. Geomechanical analyses focused on the stability of the ground surrounding the potential cavity and the influence of its geometrical parameters of it (e.g., size, shape), ground strength and pore pressure [11]. Craig [12] and Abdulla and Goodings [13] investigated the stability of soils located over cavities using laboratory centrifuge models. Augarde et al. [14] used an axisymmetric model and limit equilibrium analyses to set the stability of the potential cavity; this was expressed as the difference between the total stress in the contours of the cavity and the internal pressure divided by the material undrained shear strength. A boundary equilibrium analysis of tunnels in rigid-plastic soils was considered by Davis et al. [15]; such authors assumed that this is a similar behavior to altered ground domes in karstified rock masses. Advanced numerical models were also applied to investigate the phenomenon. For instance, Drumm and Yang [16] used the finite element method (FEM) to study the overlying material above cavities and its susceptibility to produce a ground collapse. Perrotti et al. [17] used FEM to develop charts to assess the stability of karstic cavities based on their geometrical parameters and the strength properties of the rock material. A recent work developed by Duan et al. [18] used smooth particle hydrodynamics and FEM to study the influence of karstic cavities on rock-blasting processes.

In terms of techniques used in urban areas for identifying sinkholes [1,6,19–21], the common practice includes explorations by boreholes and non-invasive techniques such as geophysical methods, e.g., electrical resistivity tomography, ground penetration radar and seismic refraction. In some cases, trenching is also used [22]. Regarding mitigation measures for sinkholes affecting buildings and infrastructures, injection of cement-based grouts or chemical grouts (e.g., polyurethane foams) is fairly common. This procedure enables filling the cavities and sealing joints and fissures while strengthening the ground [2,8,10].

Sinkhole hazards have a great economic impact in Spain, as shown by some recent works dealing with the identification and investigation of sinkholes in the Iberian Peninsula [20,23,24]. Particularly, the city of Zaragoza (NE Spain) and its surroundings are considered the highest sinkhole risk area in Europe [25]. The abundance of gypsum is the origin of such sinkholes, although the interstratal dissolution of halite and glauberite beds also contributes to their development [26,27]. In such context, this paper presents a case study in Alcalá de Ebro, a town located in a high-risk Spanish area [28]. The paper shows how to address the mitigation of this phenomenon using different techniques depending on the size of the sinkhole, the objectives pursued and the element affected. An area around a protective flood embankment built to defend the town against potential flooding from the Ebro River is analyzed. In this area, numerous sinkholes were observed. Those sinkholes caused serious economic and social issues, including affecting the main access street to the town, preventing the population access to some areas of the town and the structural affection of dwellings and the protective flood embankment itself.

A geological-geotechnical investigation was carried out, including boreholes, field tests and cross-hole seismic tomography. Based on the geological–geotechnical analysis of the area, the use of geogrids, cement injections and polyurethane (PU) injections were proposed and applied to solve the problem. A FE model was performed to evaluate the possible impact that the presence of voids (due to karst dissolution) can have on the embankment stability after conducting the injection works. It should be noted that ensuring the stability of the embankment is vital since, otherwise, a strong flood event may have catastrophic consequences. Both empirical observance and numerical results showed that the ground treatment applied was successful in its purpose of filling-consolidating the undermined regions, not threatening the stability of the embankment and preventing future sinkholes in the area.

## 2. Materials and Methods

### 2.1. Geographical and Geological Framework

Alcalá de Ebro is found in the NE of the Iberian Peninsula (Figure 1), about 35 km northwest of the Spanish city of Zaragoza, the fifth largest city of Spain. It is approximately halfway between Madrid and Barcelona (about 330 km from each one). Geologically, it is located in the sedimentary basin of the Ebro River and surrounded by three mountain systems: Pyrenees, Iberian Mountain Range and Coastal-Catalan Mountain Range. The stratigraphy of the area consists mainly of a Tertiary rock substratum of marls and gypsum, strongly altered in its superficial levels, on which Quaternary materials are deposited of variable thickness and compactness. The gypsum presents a rather monotonous aspect, being constituted by white gypsum with nodular structure. Gray-greenish marls and shales appear associated with the gypsum materials or alternating with them [28,29]. Quaternary formations outcrop widely throughout the area, arranged in several terrace levels associated with the Ebro River, as well as different glacial deposits and colluviums [29].

Alcalá de Ebro is a small town on the banks of the Ebro River, the largest river of the Iberian Peninsula, and it collects a large part of the meltwater from the southern slopes of the Pyrenees. Thus, Alcalá de Ebro is located in the Ebro Corridor, a 6-km-width corridor defined by a succession of low and medium terraces on the right bank of the river that, in a staggered manner, makes the relief descend from heights of about 300 m to 215 m at the current riverbed. The urban center is found between heights 221 m and 226 m, occupying the lowest terrace (considered the current flood plain). Here, the river adopts a meandering geometry, according to its low longitudinal slope (around 0.06%), and constitutes the regional base level [29].

In 1957 the river naturally changed its course, flooding part of Alcalá de Ebro. This led to the construction of different breakwaters and walls, creating in 1982 a protective flood embankment of more than 2 km long. Besides, urbanization and agricultural development over time led to the occupation of a ravine (Juan Gastón ravine) whose basin covers a length of 31.8 km and an area of about 232 km$^2$. Such a ravine, which opens a partially dissymmetrical valley in the Tertiary materials of the Ebro Basin, is one of the natural drainages to the Ebro River, and its elimination currently entails the existence of underground inflows through the terrain. Both this phenomenon and the proximity of the Ebro River may be the fundamental factors that condition the problem of sinkholes and karst subsidence that affect the area under study.

### 2.2. Sinkhole Occurrence and Emergency Measures

In April 2013, the area around the river's protective embankment showed soft spots on the sidewalks and pavements near some houses (Figure 2). Once those elements were removed, the existence of a large sinkhole of a surface between 35 and 40 m$^2$ and 4 m deep was confirmed. The sinkhole reached the water table and caused the foundations of the buildings to be cantilevered over the hollow. Affected building constructions were made of load-bearing walls and shallow strip foundations and showed some cracks in the main

façade forming unloading arches towards the ends. The foundations appeared cracked at their center, next to the point where the center of the collapse was located in the street.

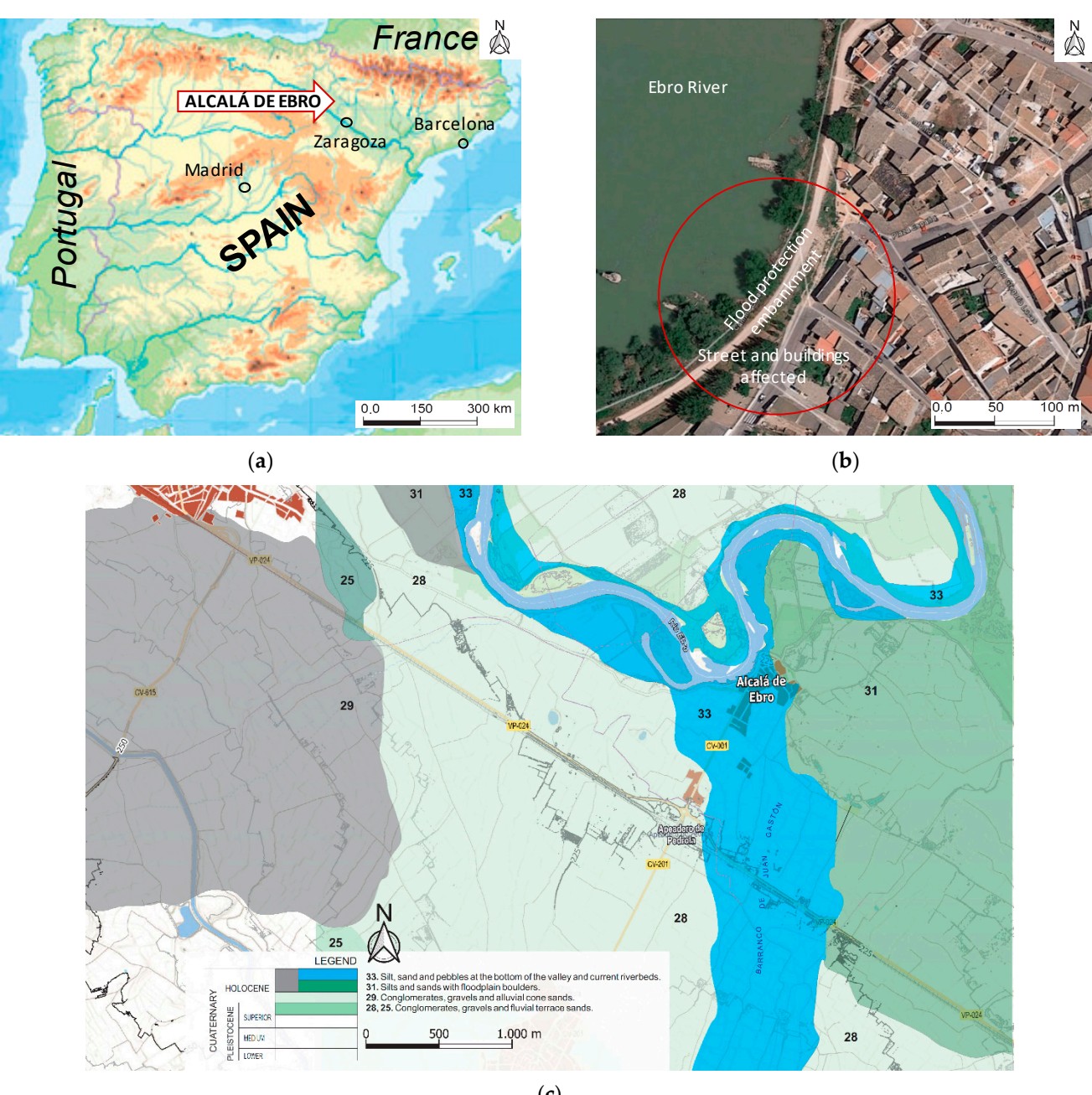

**Figure 1.** Geographical and geological framework: (**a**) Location of Alcalá de Ebro in the Iberian Peninsula; (**b**) Aerial view of the area under study (source: google maps); (**c**) Geological map of Alcalá de Ebro and its surroundings [29].

As an emergency measure and to avoid further structural deterioration, the sinkhole was backfilled with riprap blocks with a transition to gravel and pebbles at the top, trying to fill the gap below the overhang created in the foundations. Additionally, the foundations were underpinned using expansive polyurethane (PU) resins injected from the outside of the house. In the most affected façade, the number of injections was increased, also injecting them from inside the house. The walls of the adjoining buildings affected by the differential settlement were also reinforced using the same technique (PU resins).

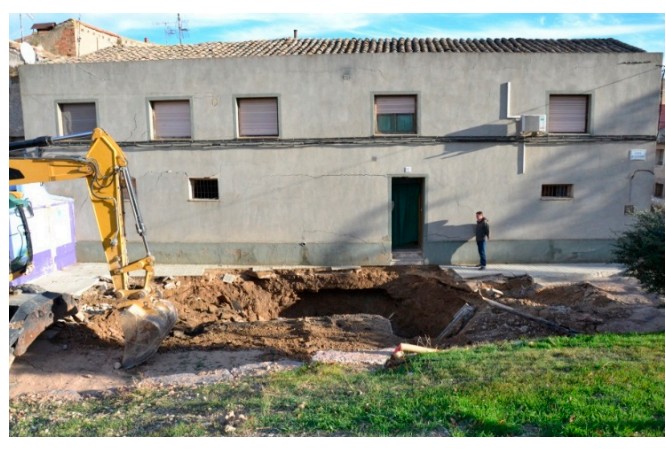

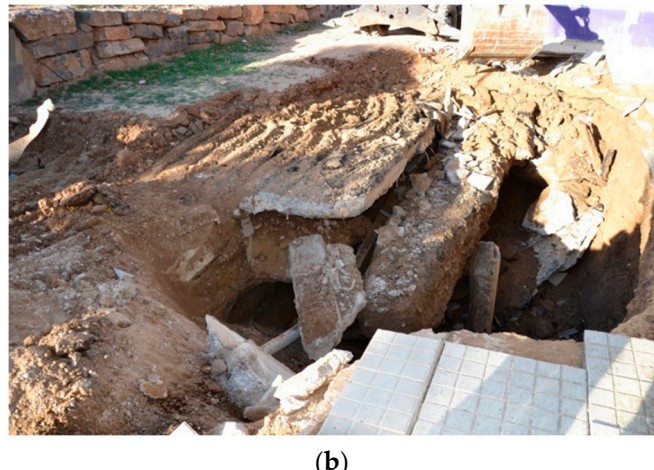

(**a**)

(**b**)

**Figure 2.** Sinkhole appeared in 2013: (**a**) General view once the pavement was removed (note foundations cantilevered over the hole between 4 m and 5 m); (**b**) Detail view showing debris from the sidewalk and pavements of the street poured into the hole.

### 2.3. Geotechnical Investigation and Geotechnical Profile

As subsidences are common in Alcalá de Ebro, some geological exploration works were performed by public services in the past [28,29], providing abundant information on the subsoil (Table 1). A complimentary geotechnical investigation (Figure 3) was carried out during the different works involved in this work, including a total of 19 Dynamic Penetration Super Heavy (DPSH) in situ tests and 4 boreholes reaching depths up to 30 m. Between those boreholes, cross-hole seismic tomography was performed. This geophysical technique estimates the value of P- and S-waves velocities of the ground, enabling identifying punctual anomalies, possible altered zones and contacts between the different geotechnical units of the ground.

**Table 1.** Geotechnical surveys and consolidation works.

| Dates | Geotechnical Surveys | Consolidation Works |
|---|---|---|
| January 2019 | 5 DPSH tests | - |
| November 2018 | 4 DPSH tests | Low mobility mortar injections and high tensile strength geogrids at embankment |
| March 2018 | 2 boreholes | - |
| December 2016 | - | Low mobility mortar injections and high tensile strength geogrids at the street |
| November 2015 | - | PU injections at the embankment |
| July 2015 | 3 DPSH tests; 2 boreholes | - |
| November 2013 | - | PU injections at the street and house |
| April 2013 | 7 DPSH tests | Emergency measure (filling sinkhole at house with riprap blocks, gravel and pebbles) |
| January 2012 | Ground penetration radar | - |
| March 2008 | 4 boreholes | - |
| September 2007 | 10 drills | - |
| 1998–1999 | 32 Dynamic penetrometers | - |
| 1995 | Microgravimetry | - |
| 1979–1980 | 7 boreholes | - |

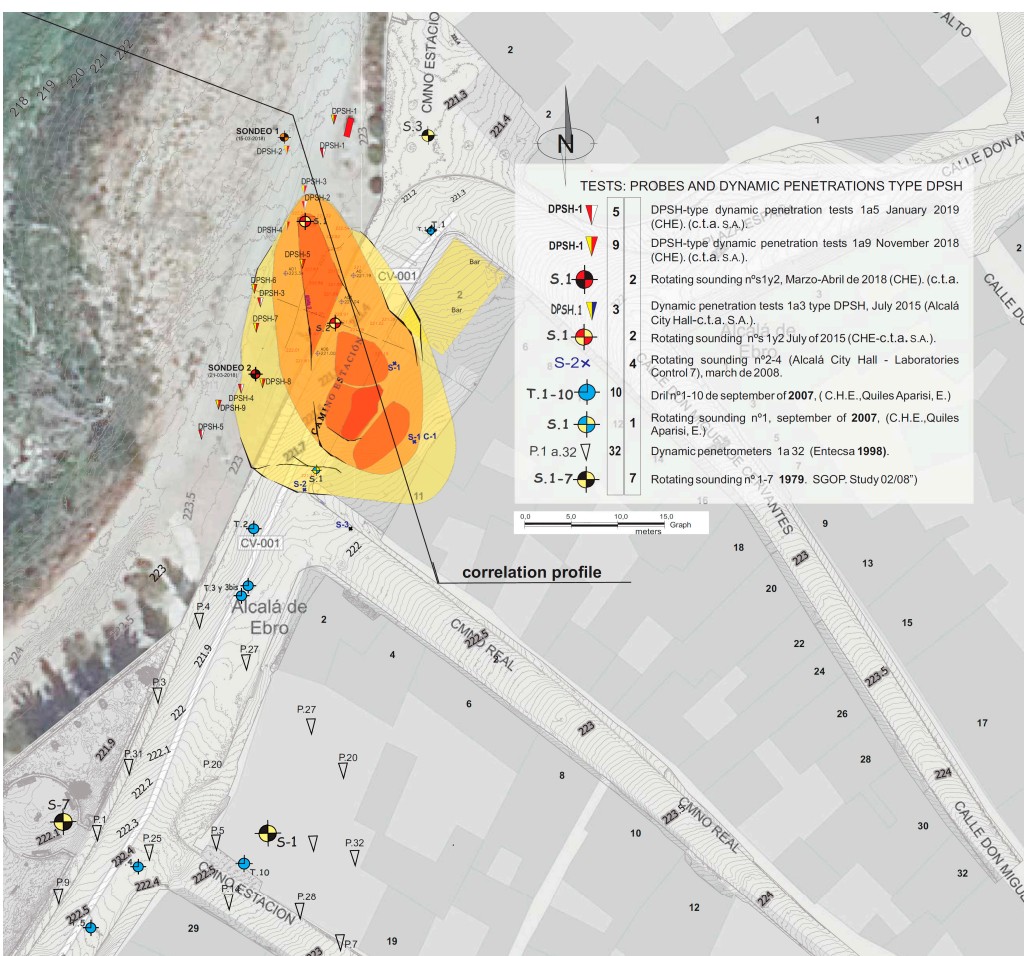

**Figure 3.** Location of the geotechnical investigation works carried out. The areas marked in red indicate sinkholes occurred in the last 15 years. The areas in orange and yellow indicate the contour zones affected by cracks in pavements and buildings, which also show evident signs of subsidence.

The use of the previous data from the literature as well as the one obtained from the new boreholes and using the cross-hole seismic technique, enabled the building of a geotechnical profile of the ground.

### 2.4. Expansive Polyurethane Resins Injections

In November 2013 (Table 1), the subsidence problem was tried to be solved by injecting expansive polyurethane (PU) resins (Figure 4) under the buildings affected and below the street itself, on the closest side to the riverbank. Such injections bind the soil by infiltrating through the porosity of the material itself and the existing voids or fissures while exerting a consolidating/compacting effect in its expansion process (their volume increases between 10 and 30 times). This injected material expands and solidifies in a very short time (between 12 s and 15 s), so it normally is located less than 2 m below the injection point. PU resins help create a good support interface, even when foundations are degraded or cracked.

A great number of injection points were applied in the street and under the buildings so that the treatment was as homogeneous as possible. Since the use of the injected elastic material neither results in a differential rigid element that could cause future settlement problems nor in substantially increasing the loads on the ground, no other special structural repairs or geotechnical treatments were conducted.

Additionally, two years later, in November 2015, expansive resin PU injections were performed in the body of the Ebro River flood protection embankment (Table 1). This action involved a total length of 30 m (Figure 4c,d) and was motivated due to the observance of

settlements at the embankment and the appearance of cracks at its surface. A length of 12 m was identified as the affected sinkhole area. Two lengths of 8 m and 5 m were defined at each side as a peripheral zone of influence, while an additional length of 5 m was defined towards the south as this area showed signs of apparent subsidence.

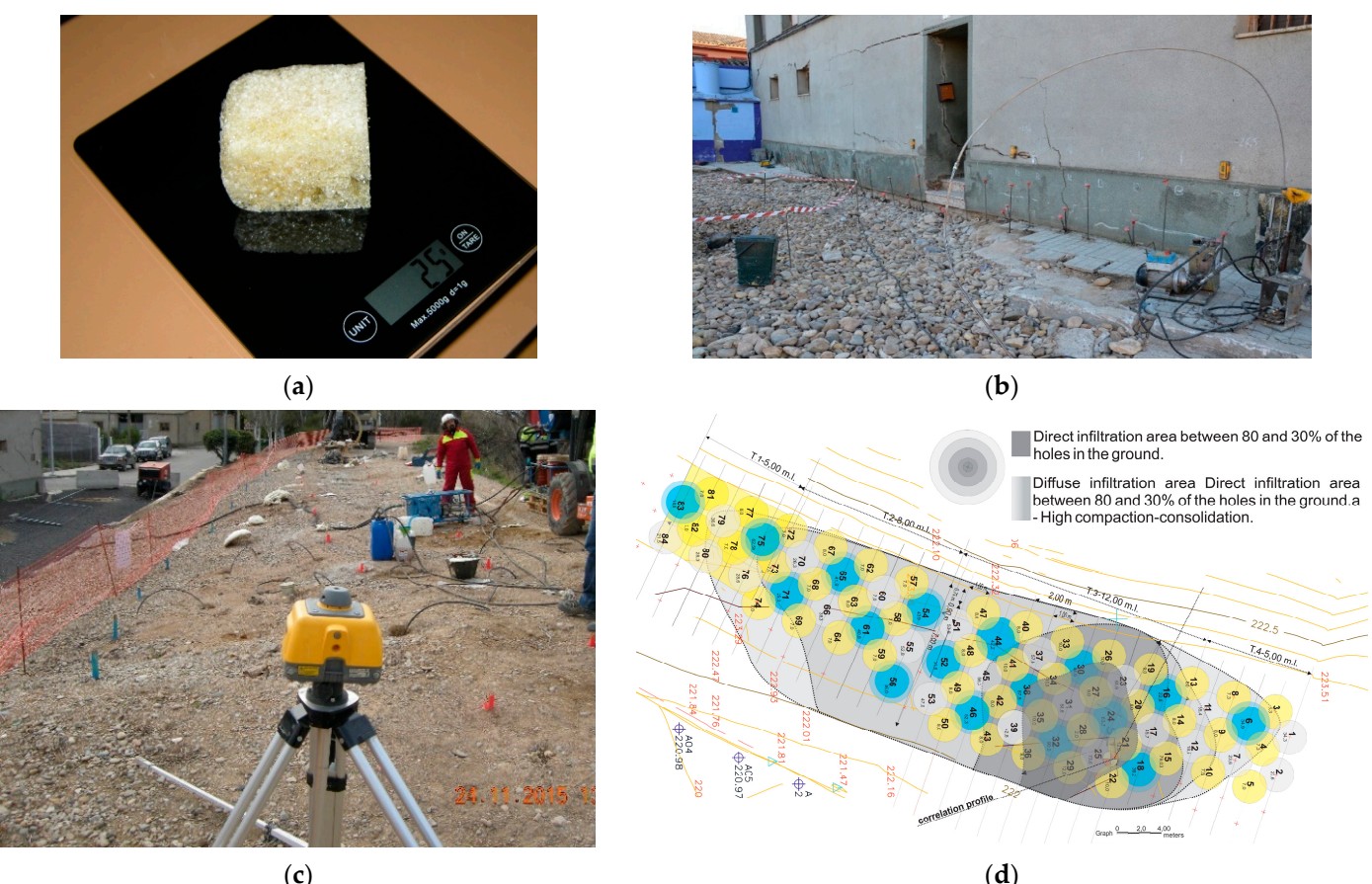

**Figure 4.** Expansive PU injections: (**a**) Resin sample (considering the weight/volume ratio, the expansion is more than 20 times the initial volume); (**b**) Façade of the building where resin injections were performed; (**c**) Injection works at the flood protection embankment; (**d**) Resin injections layout in the embankment (mesh 1 in yellow, mesh 2 in gray, mesh 3 in blue); this graph-summary has been made by superimposing all the corresponding data with the excess injection recorded. With this, a scheme is obtained in which the most sensitive area to subsidence-collapse processes is shown.

The distribution of those injections in the plan was carried out according to the geotechnical profile previously obtained and the degree of affection of the settlements observed. Three meshes were defined: mesh 1, attaining a relative depth of 5 m (up to 6 m effective); mesh 2, attaining a relative depth of 9 m (up to 6 m effective); and mesh 3, attaining a relative depth of 15 m (up to 16 m effective).

## 2.5. Low Mobility Mortar Injections and High Tensile Strength Geogrids

In December 2016 (Table 1), a total of 28 mortar injections were used to solve the subsidence problems affecting the street between the embankment and the buildings affected (Figure 5a), where PU injections were previously performed. First, an excavation and sanitation of the street and the annexed area where ground collapses were conducted until reaching a depth of 1.5 m. Then, a series of mortar injections were performed in alternating rows (staggered rows). Mortar injections help in the compaction of the ground by displacement due to the high-pressure attained and generate firm support in the form of columnar structures. Injections were made of a low mobility mortar (8.1% cement, 79.8% sand 0/2, 11.9% fly ash and 0.2% additives, approximate water/mortar ratio of 16%) and

reached a depth between 18 and 21 m. Each drilling was found to penetrate between 2 and 3 m into the intact rock substratum. After executing the mortar injections, high tensile strength geogrids were placed on the base of the excavation to reinforce it and help in tying all the injections (Figure 5b). The street at its top was thereafter reconstructed, resting on the injected zone.

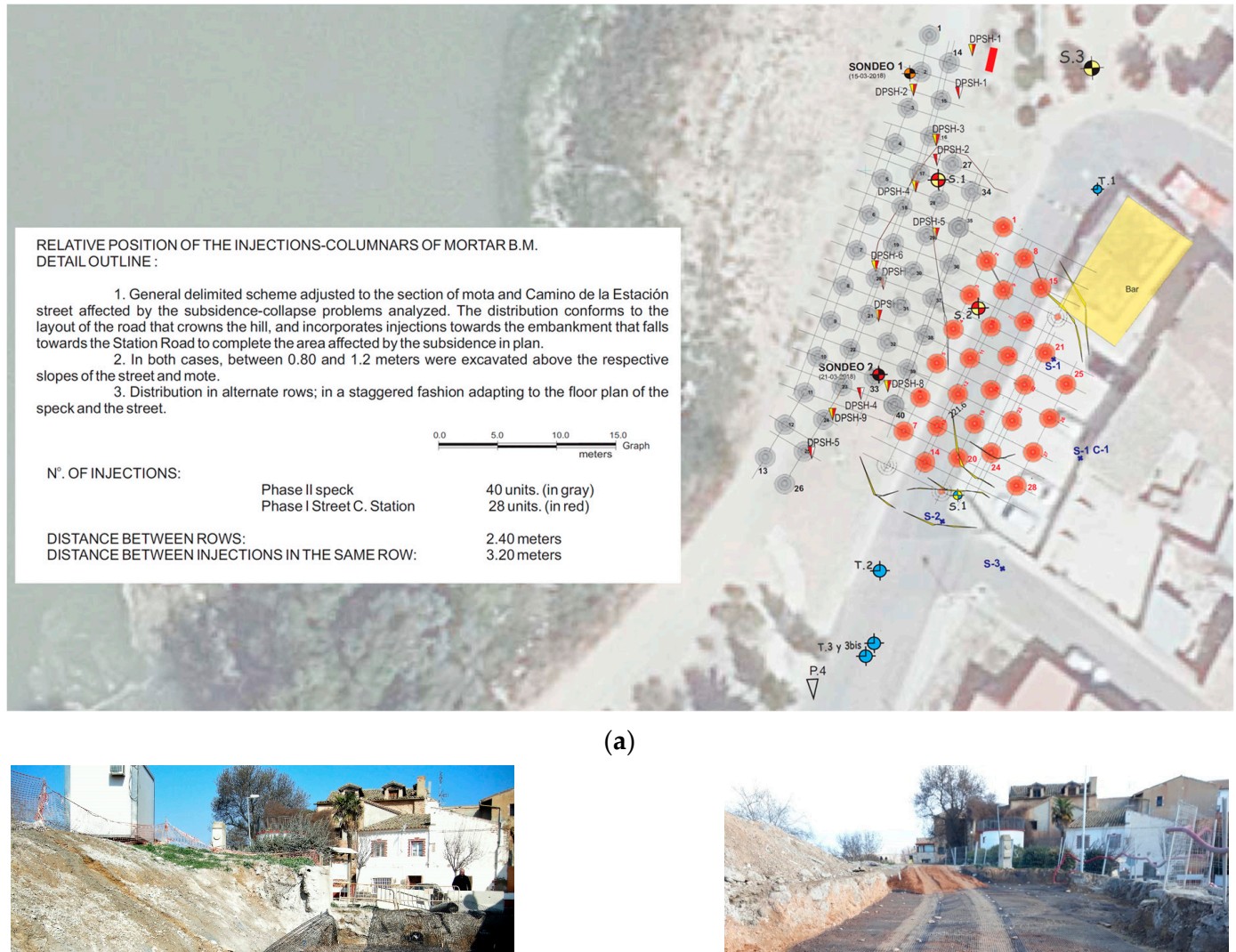

**Figure 5.** Low mobility mortar injections and high tensile strength geogrids execution: (**a**) Injections layout (in red, injections conducted in December 2016; in gray, injections conducted on November 2018); (**b**) Street area; (**c**) Embankment area.

As the subsidence process could be reactivated due to variations in the position of the water table, the Ebro River flood protection embankment was subjected in November 2018 (Table 1) to a ground treatment similar to the one applied a year before at the street. An excavation of around 1.5 m was done at the embankment, and a total of 40 mortar injections (same characteristics as the previous ones) were performed in the body of the

Ebro River flood protection embankment (Figure 5a). Those injections reached 24 m (where drilling resistance was high), and their elevation was 1 m above the surface of the street. Similar to the previous mortar injections, each drilling was found to penetrate between 2 and 3 m into the intact rock substratum. Once the injections were finished, high-tensile strength geogrids were installed at the level where the injections were performed, and the embankment was reconstructed (Figure 5c).

### 2.6. Advance Numerical Model

The commercial software PHASE2 v8.0 was used to develop a finite element analysis of the area under study. The geometry of the model corresponded to a cross-section of the embankment and the street, also including the Ebro River bank. Aside from the ground self-weight, two loads were considered: one representing the water column of the Ebro River, corresponding to the avenue that occurred in March 2015, which reached 5.7 m from the river bottom, and one representing the vehicle traffic at the street, with a uniform value of 10 kN. Boundary conditions were set as usual: fixed at the bottom and rollers at sides. Enough distance was left between the main area modeled and the boundaries to avoid any influence of them. This was set after a trial-and-error process.

Two models were built. The first model considered the ground as a homogeneous marl-gypsum rocky substrate, defined by the Mohr–Coulomb behavior (properties are found in Table 2). A circular cavity of 5.3 m in diameter was introduced below the embankment simulating the possibility of the development of a sinkhole in that area, with an overburden thickness equal to 16 m (both the diameter of the cavity and the overburden thickness were based on the results obtained from the geotechnical investigation).

**Table 2.** Ground material properties.

| Material | Unit Weight (kN/m³) | Young Modulus (MPa) | Poisson Ratio (–) | Tensile Strength (kPa) | Cohesion (kPa) | Friction Angle (°) | Permeability (m/s) |
|---|---|---|---|---|---|---|---|
| Embankment | 20 | 16 | 0.3 | 0 | 5.0 | 32 | 0.001 |
| Artificial filling | 19 | 7 | 0.3 | 0 | 9.8 | 15 | 0.001 |
| Marl-gypsum substrate | 22 | 36 | 0.3 | 250 | 96.1 | 33 | $10^{-7}$ |
| Alluvial material | 20 | 16 | 0.3 | 0 | 31.4 | 20 | 0.01 |
| Material with resin injections | 14.2 | 20 | 0.3 | 0 | 6.5 | 41 | $10^{-8}$ |

The second model considered the ground following the geotechnical profile obtained in this work (see Figure 6 and Table 2 for their corresponding properties). The level found between a depth of 16 m and 21 m (from the embankment) was considered the location of the alteration and dissolution of the marl–gypsum rocky substrate, so one cavity of 5.3 m in diameter was located at various positions.

Additionally, the deterioration of the karstic level was introduced as a progressive variable, evolving by incorporating increasingly unfavorable geotechnical parameters based on the common shear strength reduction technique usually used for computing the safety factor in finite element analyses.

In both models, consolidation operations done before 2018 were considered, i.e., resin injections in the embankment done in 2015, mortar injections performed in the street in late 2016 (along with the installation of the high tensile strength geogrids) and an embankment regrowth of the embankment performed in 2017 (performed due to flooding issues and not directly related with the karstic phenomenon). Resin injections were treated as a "new" ground material (see Table 2). Mortar injections and geogrids were modeled as linear elastic materials with a Young modulus of 10 GPa and 2000 kPa, respectively.

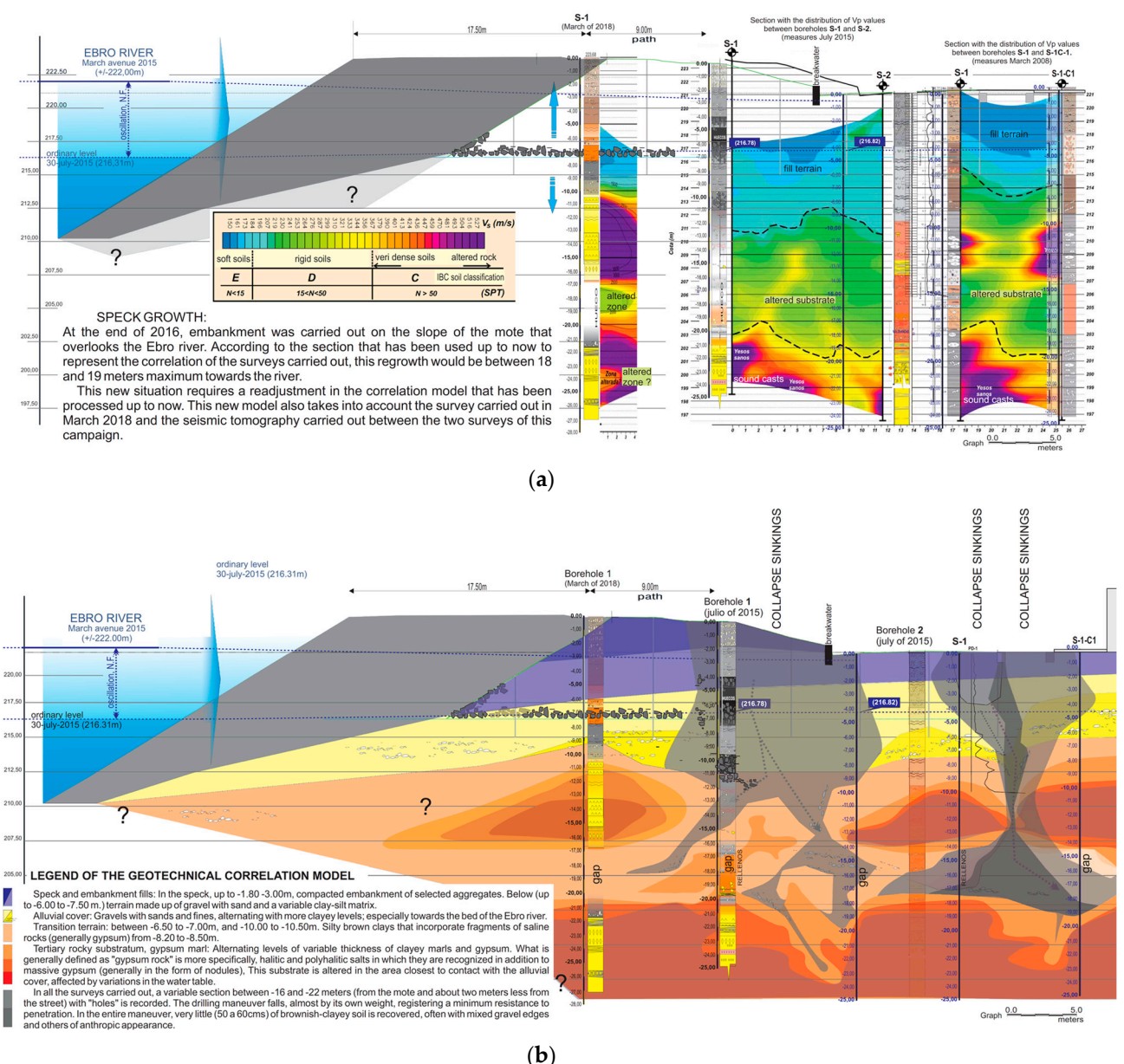

**Figure 6.** Geotechnical investigation results with geological interpretation: (**a**) Correlation between boreholes S-1 and S1-C1 (March 2008), S-1 and S2 (July 2015) and S-1 and S-2 (March 2018), including the seismic tomography between them and the DPSH tests performed; (**b**) Proposed geotechnical profile following section A-A' (see Figure 3).

## 3. Results and Discussion

### 3.1. Geotechnical Profile

From the results of the boreholes, in situ test and cross-hole seismic tomography tests conducted, the stratigraphy of the area under study (Figure 6) may be divided into four main units: (i) the filling of the embankment, which consists of gravels with sands and a variable brown silty clay matrix, although up to a depth between 1.8 m and 3.0 m compacted gravels prevail; (ii) alluvial cover, composed of gravels with sands and fines, alternating with brown silts and clays, often with traces of organic matter, towards the Ebro River, and with a chaotic appearance due to being affected by the karstic processes; (iii) a transition ground, consisting of brown silty clays that incorporate fragments of evaporitic rocks (generally gypsum) and (iv) tertiary rock substratum. Deep boreholes indicate that this substratum can be divided into four lithostratigraphic units, in ascending order [30–32]:

(1) marl and anhydrite basal unit; (2) halite unit; (3) glauberite–halite unit; and (4) anhydrite unit. The Alcalá sinkhole, according to its elevation (221 m.a.s.l.), is directly underlain either by the lower part of the anhydrite unit or the upper part of the glauberite-halite unit [28].

This last unit is altered in its upper part and is affected by the variations of the water table. Consequently, internal dissolution processes are generated, facilitating the formation of sinkholes. In all the boreholes conducted, between a depth of 16 m and 22 m (measured from the top of the embankment, about 2 m less from the street), several holes were recorded, with minimum resistance to penetration (drilling maneuver falls almost by its own weight), only recovering about 50 cm of brown silty clay soil with gravel pebbles. This means that the origin of the sinkhole problems is located at a level of approximately 6 m of thickness, about 14 m below the street, formed by very soluble rocks (gypsum in the form of nodules, glauberite, halite, thenardites and epsomites [28,31,32]). Materials (gravels with sands) coming from the detrital levels located above or from anthropic ones are found here, filling the holes left by karstic dissolution processes.

This dissolution process results in subsidence rising towards the surface, which produces sinkholes. The phenomenon is related to the variations in the water table: sinkholes normally appear between June and October, being that period when the water level at the river is the lowest [33] and the contributions from the headwaters of the catchment basin is low.

### 3.2. Expansive PU Resins Injections

Injection of expansive resins under the buildings affected, performed in 2013, showed that the procedure was quite effective, recording minimal settlement and movements according to the topographic control. However, the injected area in the street was not enough. Relative subsidence in the center of the area affected by a sinkhole of about 12 to 15 cm was observed (Figure 7). This is explained by the depth of the injection in the street area, of only 4 m. Considering the extent of the apparently sunken area (more than 20 m) and the depth of the area affected by dissolution processes (between 14 and 20 m), this depth was clearly inadequate.

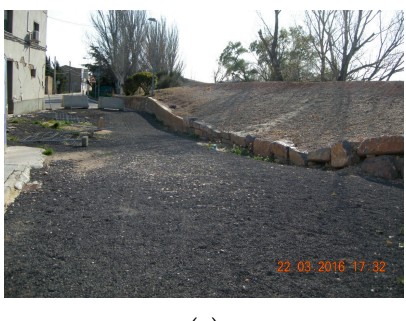

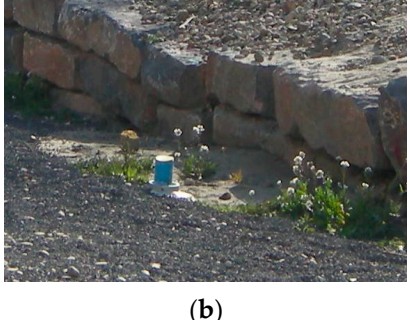

(**a**)        (**b**)

**Figure 7.** Settlement observed in the street after performing the PU resin injection treatment: (**a**) general view; (**b**) piezometric pipe (corresponding to borehole #2) used as a reference point; this borehole had a depth of 25 m and the pipe rested on firm ground. Note how the relative descent of the surrounding ground causes the pipe to protrude from the ground.

Regarding resin injections performed in the protection embankment in 2015, overconsumption reaching more than 80% was recorded in some areas below the embankment. This was especially observed at a depth between 9 and 15 m, being those areas (Figure 7) were the ones where cracks were seen at the surface and where the highest differential settlements were recorded. This indicates the presence of voids or strongly undermined areas. Considering the extent of the apparently sunken zone (more than 20 m) and the depth of the zone affected by dissolution and undermining processes, the treatment resulted in being insufficient. In this case, the main limiting factor was the injection method and equipment itself, which had many difficulties in reaching high depths before the resins started to solidify in the tubes.

The use of expansive PU resin injection [1,28] showed to be useful as a simple and fast method to control and delay the sinkhole phenomenon. This technique induced minimal settlement and movement issues in the buildings. Due to the great extent of the problem, the technique was not enough to totally control the sinkhole phenomenon: late subsidence was observed. However, in both cases analyzed (buildings affected and street), the PU resin injections generated an advantageous effect: due to the flexibility given by the resins, the dissolution processes on the ground tended to generate a "general bending" effect at the surface more than a sudden sinkhole (collapse) phenomenon. At least, this effect delayed the collapse. In fact, after the injection of expansive PU resins, a slight decrease in subsidence rates was observed in all the areas under study.

### 3.3. Mortar Injections

As intakes were expected to be high due to the existence of voids or low consolidated zones between 16 m and 21 m, mortar injections were performed to reach a minimum pressure of 20 bars or until injecting 500 L of mortar per every 0.5 m.

Mortar consumption for the first injection phase (December 2016), performed on the street, reached an average value of 200 L/m. The combination of such low-mobility mortar columns [1,28] with the installation of high-tensile strength geogrids [34] created a reinforced membrane at the top of the injections, which turned into the foundation of the street granular layers. That ensured the future stability of the street, which did not show any evidence of significant settlements during the next four years.

Mortar consumption for the second injection phase (December 2018), performed at the embankment, showed higher values than the previous ones. Here, some points reach consumptions of more than 400 L/m in the closest row of injections towards the river. That values are exceptionally high and indicate the presence of voids in the ground. In addition, half of the mortar intakes, around 35 kL, were concentrated in the stratum identified as the origin of sinkholes, i.e., between 14 m and 21 m in depth. Conversely, a reduced intake was observed in the rest of the rows.

That suggests the most problematic areas are those next to the Ebro River. Thus, both fluctuations in the river level (which means fluctuations of the water table) and the underground inflows coming from the old surficial ravines (today closed) are the most plausible factors conditioning sinkholes in the area under study. There is no evidence that the origin of the sinkhole phenomenon is produced by the water discharge through the alluvial cover, as is the case in other case studies [23]. In the present case study, the Ebro River is the main actor.

Mortar injections showed to be effective in controlling the sinkhole phenomenon. Precision leveling was carried out since the treated area and its surroundings showed that the surface was stable. Both in the embankment and in the street, no deformations were recorded that indicate any subsidence or sinking process is still active.

### 3.4. Advanced Numerical Model

As can be observed, in the first model (Figure 8), the existence of the cavity means a maximum displacement at the surface of about 0.076 m, while this value was nearly 0.115 m at the cavity crest. In the second model, where the ground was considered according to the geotechnical profile seen in Section 3.1 (Figure 6), the most unfavorable position of the cavity was found to be below the embankment. In this model, for the strength properties without any reduction, maximum displacement at the surface was found at around 50 mm, this value being nearly 18 mm at the cavity crest, while the safety factor was 1.2. When the strength reduction factor technique was applied to the model, progressively reducing the strength properties of the geological materials, a minimum safety factor of 1.05 was obtained (Figure 9). Maximum displacement at the surface was around 76 mm, this value being nearly 112 mm at the cavity crest. As the safety factor was greater than 1.0, the cavity was stable against collapse, meaning that the mortar injections and geogrids treatment applied were capable of successfully controlling the sinkhole phenomenon.

## TOTAL DISPLACEMENTS:

ANALYTICAL SUMMARY

| PHASE | INTERNAL PRESURE (Kpa) | EQUIVALENT LOAD FACTOR | TOTAL DISPLACEMENT CALVE (m) | TOTAL PERPENDICULAR DISPLACEMENT (m) | TOTAL DISPLACEMENT ON SURFACE (m) | SECURITY FACTOR |
|---|---|---|---|---|---|---|
| 1 | 148.400 | 0.2780 | 0.018 | 0.050 | 0.050 | 1.800 |
| 2 | 92.270 | 0.1730 | 0.024 | 0.050 | 0.055 | 1.350 |
| 3 | 83.210 | 0.1560 | 0.026 | 0.051 | 0.056 | 1.340 |
| 4 | 52.080 | 0.0976 | 0.038 | 0.061 | 0.060 | 1.350 |
| 5 | 42.670 | 0.0800 | 0.044 | 0.066 | 0.062 | 1.360 |
| 6 | 26.670 | 0.0500 | 0.055 | 0.075 | 0.065 | 1.250 |
| 7 | 10.670 | 0.0200 | 0.071 | 0.087 | 0.069 | 1.040 |
| 8 | 5.330 | 0.0100 | 0.085 | 0.098 | 0.072 | 0.980 |
| 9 | 5.180 | 0.0097 | 0.097 | 0.105 | 0.074 | 0.980 |
| 10 | 0.000 | 0.0000 | 0.112 | 0.114 | 0.076 | 0.900 |

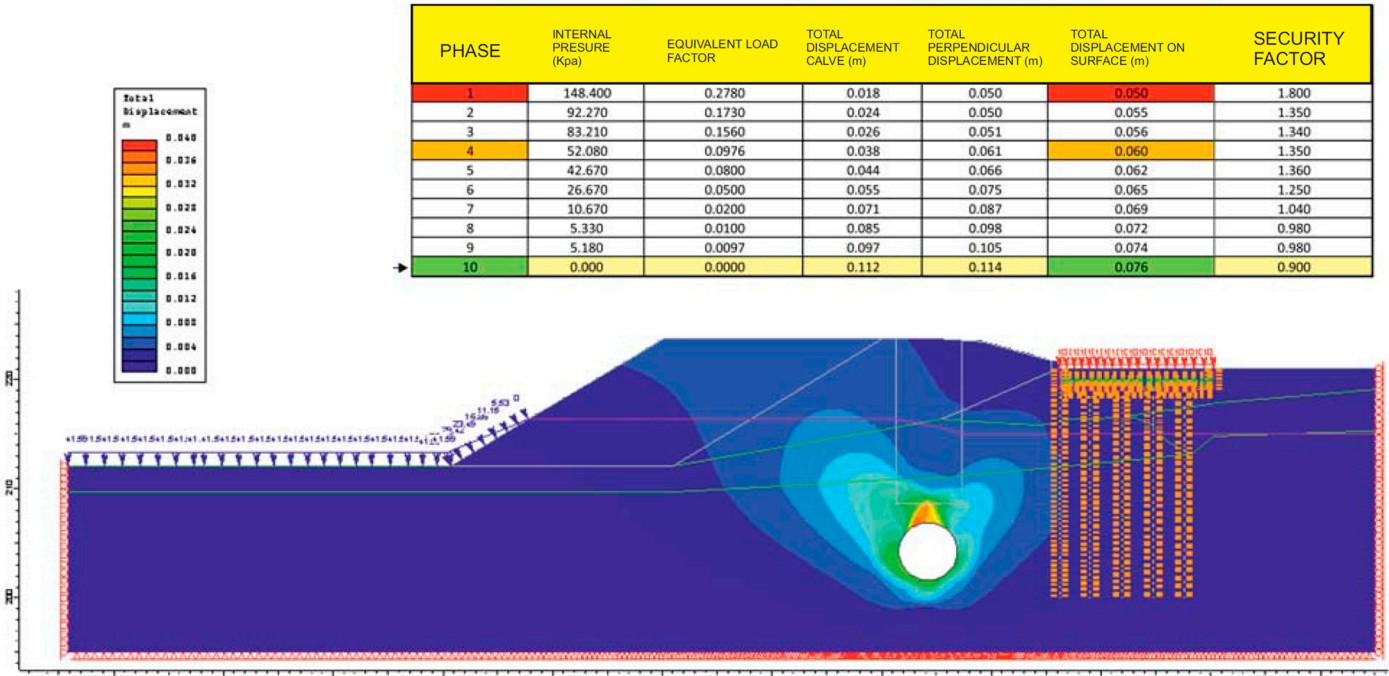

**Figure 8.** Final phase of the analysis with a homogenous substratum and a fixed position of the cavity in the model.

## TOTAL DISPLACEMENTS:

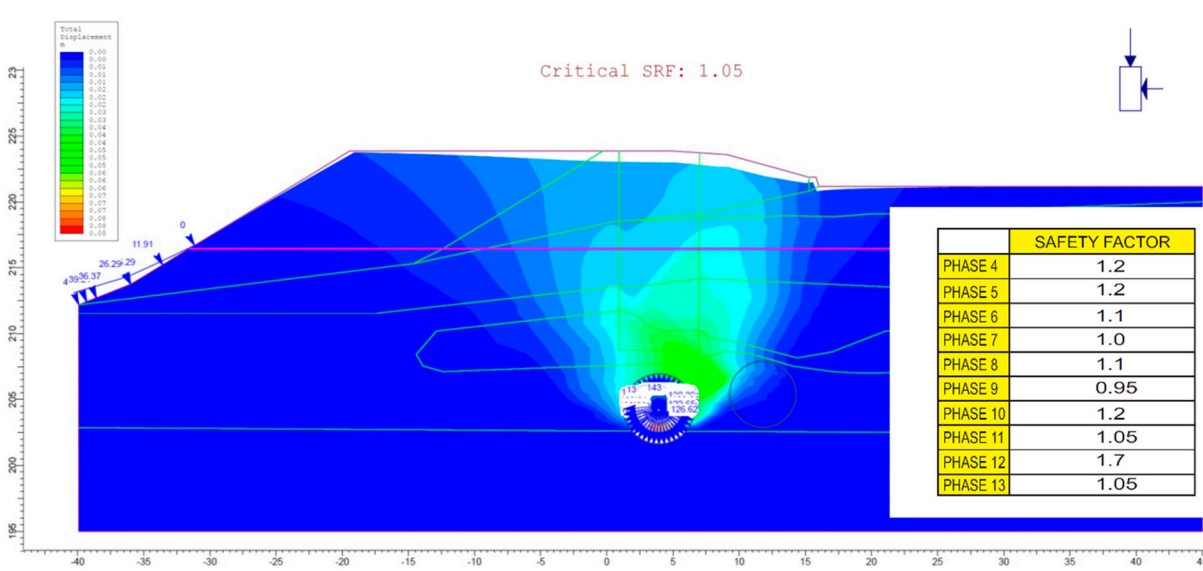

| | SAFETY FACTOR |
|---|---|
| PHASE 4 | 1.2 |
| PHASE 5 | 1.2 |
| PHASE 6 | 1.1 |
| PHASE 7 | 1.0 |
| PHASE 8 | 1.1 |
| PHASE 9 | 0.95 |
| PHASE 10 | 1.2 |
| PHASE 11 | 1.05 |
| PHASE 12 | 1.7 |
| PHASE 13 | 1.05 |

**Figure 9.** Final phase of the analysis with a non-homogenous substratum according to the geotechnical model and the most unfavorable position of a potential cavity.

It should be noted that the stability of the area needs the cavity to remain submerged with sufficient internal water pressure. This aspect is expected to be always fulfilled since the cavity is connected to the alluvial aquifer, and the position of its crest with respect to the water table leads the cavity to never empty. However, the natural progressive evolution by the dissolution of the karstic ground and the existence of future summer periods of extremely low water in the Ebro River triggered by climatic change [33], which matches the

cut off of the irrigation of the farms that drain into this basin, may give rise to an important change into the hydrogeological conditions, thus reactivating the sinkhole.

## 4. Conclusions

The problems caused by sinkholes in the Ebro River valley due to evaporitic karstic materials are becoming more and more worrying every day, with buildings and communication routes being affected. The traditional idea that this is a natural and unpredictable phenomenon has shown not to be true, and in most cases, the origin or main cause is related to human actions that trigger or accelerate processes that otherwise would not occur or have a much slower development.

Some of these actions include diverting or impeding watercourses, performing fillings in old watercourses and digging wells that connect superficial aquifers with deep saline levels. These actions produce radical changes in the surface and sub-surface hydraulic regimes, worsened by the existing irrigation systems that also contribute to changes in the hydrogeological conditions. In Alcalá de Ebro, nearly all those issues make an appearance and are combined, resulting in sinkholes affecting the population. Sinkholes have been common in the area for many centuries, and in recent years, they affected the southwestern part of the town.

Geotechnical investigations involving boreholes, in situ tests and geophysical techniques are needed to identify the origin, extent and probable evolution of sinkholes. This helps elaborate geotechnical profiles that make it possible to understand the sinkhole process and propose ground treatments. In Alcalá de Ebro, the origin of sinkholes was set in a stratum of 4 to 6 m of thickness, made of an alternation of very soluble rocks (gypsum, glauberite and halite, among others) and located at a depth of around 15 m from the surface. At this stratum, boreholes found holes and fillings coming from alluvial detrital levels.

Different mitigation techniques were shown and applied in this case study. Basic approaches to solving the problem, such as filling the open holes with riprap and rubble (sometimes even with concrete), are normally used as emergency measures. They can help in keeping the stability and avoid cantilevered schemes of foundation elements such as slabs and strip foundations. However, they showed in this case study not to correct the main problem. Sinkholes reappeared again after some years.

Expansive PU resin injections are unable to control the sinkhole phenomenon in large areas. However, it is a fast and economical technique that helps the ground to behave more flexibly, thus reducing and delaying the possibility of a ground collapse. PU resins also showed not to be useful for high depths (more than 15 m), as they start to solidify before being injected at their target position.

Low mortar mobility injections combined with high tensile strength geogrids showed to be the most effective technique for addressing the mitigation of large sinkholes. They are expensive, especially if high mortar consumption is needed (due to the existence of large voids or very low consolidation areas), but they ensure the stability of infrastructures like embankments or streets built above them.

The Ebro River was identified as the main cause of the sinkhole phenomenon. It controls the water table fluctuations as well as the underground watercourses. It also helps in ensuring the stability of the area: advanced numerical models showed that such stability needs the potential karstic cavities to remain submerged, which is guaranteed as the ground here is connected to the Ebro River alluvial aquifer. However, future changes in the hydrology and hydrogeology of the area can have a negative influence, increasing the sinkhole hazard. Further investigation in the area under study should, therefore, focus on the study of the relationship of sinkhole development with precipitation regimes, hydrogeological changes (including seasonal thaw variations) and hydrodynamic affections produced by the incipient climatic change.

**Author Contributions:** Conceptualization, A.G., F.J.T. and J.G.-R.; methodology, A.G. and M.P.-P.; software, A.G. and M.P.-P.; validation, A.G., F.J.T. and J.G.-R.; formal analysis, A.G., M.P.-P. and J.G.-R.; investigation, A.G., M.P.-P. and O.A.-P.; resources, A.G. and M.P.-P.; data curation, A.G. and

O.A.-P.; writing—original draft preparation, A.G.; writing—review and editing, F.J.T., J.G.-R. and O.A.-P.; visualization, F.J.T., J.G.-R. and O.A.-P.; supervision, J.G.-R. and F.J.T.; project administration, F.J.T. All authors have read and agreed to the published version of the manuscript.

**Funding:** This research received no external funding.

**Institutional Review Board Statement:** Not applicable.

**Informed Consent Statement:** Not applicable.

**Data Availability Statement:** The data may be available on request from the first author but not publicly available due to being private.

**Acknowledgments:** We would like to thank the City Hall of Alcalá village for providing permissions and logistic support during the measurement campaigns. Thanks are also given to the Environmental Management Area of the Ebro Hydrographic Confederation (CHE). We are also grateful to Jesús Rico (Associated Technical Consultants, CTA, S.A.P.) for his collaboration in the geotechnical reports.

**Conflicts of Interest:** The authors declare no conflict of interest.

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
