# Peer review of "Identification and Mitigation of Subsidence and Collapse Hazards in Karstic Areas: A Case Study in Alcalá de Ebro (Spain)"

_applsci, doi:10.3390/app13095687_

Round 1

Reviewer 1 Report

Dear Editor

I whoud thank you for the invitation to review the manuscript intitled "Identification and mitigation of subsidence and collapse hazards in karstic areas located in areas with evaporitic rocks" The present study discussed very important topic; identification and mitigation of a series of sinkholes appeared in the town of Alcalá de Ebro A thorough geological investigation including boreholes, field tests and geophysics. The methodology is well done and in the scope of the journal, however, major revisions are suggested.

MAJORS COMMENTS :

Novelty and scientific contribution : authors should add the novelty of your work, the discussion should be improved as possible and separate from conclusion and compared with other cased study.

Form, presentation and language: the paper seems as technical report (diagnostic and remediation) it not a scientific paper (need the prediction and p , A lot of chronological events and interventions; authors should add table summarize by year the different interventions to facilitate tracking the chronology of the investigation and diagnostic..... is very difficult for future lectors to track and understand. Moreover, the English should be improving as possible with scientific manner.

MINORS COMMENTS :

Title: I believe that it will be better authors to add the case study (study area) is good for the paper the apparition and visibility in the database.

Abstract:

Authors should add the findings and results in the abstract.

Keywords

Add the study area to the keywords and adapt the keywords with the content, some are not existing in the abstract.

Introduction:  Line 56 57

"These aspects may be aggravated by the current climate change" (put references or remove)

2.1. Geographical and geological framework: Line 129-138

Authors should add reference

Figure.1 Authors should add scale and orientation of all figures (a, b and c).

Figure.1,c: Authors should redraw with high quality and adequate legend, and the log stratigraphic should match as the text.

2.3. Geotehcnical investigation and geotechnical profile Line 168:

correct Geotehcnica orthograph

Authors should add the full expression of the abbreviation DPSH as first apparition in the text.

Figure.3 Authors should add scale and orientation.

2.4. Expansive PU resins injection

Authors should add the full expression of the abbreviation PU as first apparition in the text.

2.6. Advance numerical model

"The first model considered the ground as a homogeneous marl-gypsum rocky substrate", the suggested model it correlate with drilling hole (real stratigraphic log)?

3.1. Geotechnical profile

Line 285

authors should change saline rock by evaporitic.  

Line 286-288

Authors involved different mineralogical and Crystals (halitic and polyhalitic, glauberite crystals and thenardites), without references or X-Ray analysis to confirm this results?

In Figure. 6 it clear the unsaturated zone, water table and the direct influence of the river level fluctuation or hydrodynamic change and the Spain in the few years had a big climatic change as said in the introduction (Line 56 57); "These aspects may be aggravated by the current climate change" is the key to understand exactly what happen in your case study, authors may exploited better this relation for more insights.  

Line 301

Authors gave different mineralogical rocks (gypsum in the form of nodules, glauberite, halite, thenardites and epsomites) without references or X-Ray analysis to confirm this results.

Line 301  314

"that may give" and "The cause of such ineffective result may be put dow That may give"

were not scientific expression, please improve your English.

Line 362 364

Reformulation with scientific style.

4. Discussion and conclusions

The conclusions should be separate from the discussion and the discussion should improve as possible with similar case study and add the limitations of the proposal method (diagnostic) and the I suggest to do a relationship with the precipitation, hydrogeological, hydrodynamic, climatic change by years for more insights in the prediction of sinkholes.

Author Response

Dear reviewer, please find attached the response to your review. Thanks

Reviewer 2 Report

This paper presents the identification and mitigation of a series of sinkholes appeared in the town of Alcalá de Ebro (Spain). A thorough geological investigation including bore-20 holes, field tests and geophysics were performed. The topic of the paper is interesting and this research falls within the scope of the Journal. However, after reading carefully the manuscript, I agree to its publication in condition to Major Revision. Despite considerable works made in this study, I believe this paper lacks sufficient explanations of: (1) literature review, (2) novelty, (3) methodology and (4) results discussion. Specific reasons are stated in the following:

1. The literature review needs to be enhanced to show different causes of sinkholes appearance including type of soil, weather condition, and etc. Moreover, the literature review should be involved two main parts including causes of sinkholes appearance and approaches to avoiding sinkhole appearance.

2. The limitations of the previous studies should be highlighted in the revised paper.

3. The novelty and necessity of this paper are questionable. They should be clearly presented in the last paragraph of introduction section of the revised paper.

4. The quality of Figures 1(c), 4(d) and 6 should be improved.

5. In page 5, line 161: Please provide the type of injection performed including injection method and type of material. It should be provided in the revised paper.

6. In page 5, line 163, what type of approach was used to reinforce the wall against different settlements? It should be provided in the revised paper.

7. Figure 3 is huge. The size of this figure should be decreased and accordingly, the font size of texts and signs provided in this figure should be increased.

8. Some figures should be provided in the revised paper to show the results (value of P- and S- waves) obtained from geotechnical investigation (for the area shown in Figure 3) to help the reader to reach a deep understanding of the geotechnical condition of the region.

9. Section 2.4: Please provide more details about the Expansive PU resins and the method of its injection in the revised paper. Moreover, how many numbers of injection hole were used to inject the PU into the ground or embankment? What criteria (such as injection pressure) was used to stop the PU injection process?  All of them should be described in the revised paper.

10. Section 2.5: What of type of mortar was used for injection? What are the details of the mortar? How the mortar is injected? What is the criteria for stopping the injection process? Based on what criteria the number of holes for mortar injection is determined? All of them should be described in the revised paper.

11. The details of the finite element model developed in this study are vague. For instance, geometry of the model, type of material behaviors, model boundary conditions, type and number of elements, loadings, type of analysis, all components of the model, the interaction between all components of the model and etc. should be introduced and provided in the revised paper. Moreover, some views of the model should be shown in the revised manuscript.

12. How the short-term and long-term behaviors of soil, PU and mortar are defined in the model?

13. More discussions should be added to the revised paper to elaborate the results.

Author Response

Dear reviewer, please kindly find the attached file

Reviewer 3 Report

I 1. suggest the title as “Identification and mitigation of subsidence and collapse hazards in karstic areas”

2.       Re-write ABSTRACT (check 2nd and 4th sentence)

3.       Incomplete sentence (A complementary finite element analysis is conducted)

4.       Keywords do not show the actual characteristics of the study

5.       Strengthen the INTRODUCTION part with latest citation resource and link the literature with objectives of research. Check citation #2 and #3. Give some literature on identification techniques. Also this part is full of long and unclear phrases, revise it.

6.       Methodology Section: After field investigation, authors used PHASE computer code and presented some know results. What is new or modification? Also boundary conditions are missing.

7.       Table #1, there are 5 different geological materials, how authors considered these material in a model. Can we model material heterogeneity in PHASE simulator, if yes, then how? Provide this information in methodology section for readers. Also, the Permeability (m/s) of embankment and artificial filling is same but cohesion fore and friction do not parallel with this. I think these values are not original.

8.       Figure #6, lettering issue. Also explanation of this figure is poor

9.       Subheading 3.3 do not make any sense at current situation

10.   I think results are poorly managed and not much deep.

11.   I suggest results and discussions can be presented together.

12.   Only main outcomes of results and limitation of study can be presented in Conclusions.

13.   I did not found any new/interesting thing in this manuscript.

14.   Language is very poor. It does not meet the demands of scientific writing.

Author Response

(The authors gave the same response as above.)

Round 2

Reviewer 1 Report

Authors replied seriously to all comments and suggestions, the ms had been highly improved in this current form I believe it acceptable for publication.

Reviewer 2 Report

All comments made by the reviewer were addressed and implemented in the revised paper.

Reviewer 3 Report

Accept